# Towards Optimal LLM Selection

## Abstract

Generative AI and LLMs in particular are heavily used nowadays for various document processing tasks such as question answering and document summarization. Enterprises are incurring huge costs when operating or using LLMs for their respective use cases. In this work, we propose optimizing the usage costs of LLMs in a quality-aware manner for document summarization tasks. Specifically, we propose to exploit the variability of LLM performances across different types and formats of data to maximize the output quality while maintaining expected costs under a budget and latency within a threshold. This presents two challenges: 1) estimating the output quality of LLMs at runtime without invoking each LLM, 2) optimally allocating queries to LLMs such that the objectives are optimized and constraints are satisfied. We propose a model to predict the output quality of LLMs on text summarization, followed by an LP rounding algorithm to optimize the selection of LLMs. We study the problems both theoretically and empirically. Our methods reduce costs by $40\% - 90\%$ while improving quality by $4\% - 7\%$. In addition to the quantitative results, we further show that our model quality estimation aligns majorly with human preferences through a user study.

## 1 Introduction

Generative AI-based technologies are transforming the way we approach most tasks nowadays and have the potential to significantly disrupt the global economy. OpenAI's ChatGPT Ope (b) and other GPT-based large language models available through OpenAI web APIs, along with other open source LLMs such as LLAMA2 Touvron et al. (2023) etc. have proved tremendously successful in document processing tasks, such as question-answering and summarization. However, usage of LLMs (especially, in API-based scenarios) comes at a cost and it is important to understand the underlying economic ramifications Ashoori (2023); Sallam (2023).

In practical scenarios, different Large Language Models (LLMs) come with diverse costs and capabilities. Table 1 lists the costs associated with different Open AI-provided LLM APIs. We can see that the costs are quite varied across LLMs. Not only the costs, the capabilities of different LLMs for different tasks and different types of documents can be potentially varied, and are non-trivial to estimate. In fact, there seems to be **no clear hierarchy of models** in terms of their costs and capabilities. For instance, we have empirically observed that there is a significant difference in the summarization capabilities of GPT-3.5-Turbo and `text-davinci-003` on documents containing data in certain formats, such as tables versus lists. Predicting or estimating the output quality of LLMs for any given context and task, without actually invoking the LLMs is non-trivial and challenging. Most existing methods Chen et al. (2023); Jiang et al. (2023a) need the LLM outputs at run time to measure the output qualities. However, this can lead to increased costs and latency. The choice of metric for estimating the quality of generated texts for different tasks quantitatively is also a difficult problem, as existing metrics Lin (2004); Banerjee & Lavie (2005); Papineni et al. (2002) often do not correlate well with human perceptions of quality.

Estimating the output quality alone does not solve the problem. It is still non-trivial to determine which model a task should be directed to when **cost and latency considerations** come into the mix. There might be system-imposed budget constraints, or the user might be interested in minimizing their costs, though not at the expense of the output quality. For example, randomly routing a percentage of queries to cheaper/weaker LLMs to lower costs might end up hampering user experience. One needs to ideally find an optimal routing of queries or tasks to models to satisfy required constraints on costs, quality or latency.

| Model | Input Cost | Output Cost |
|---|---|---|
| `text-davinci-003-002` | $0.0200 / 1K tokens | $0.0200 / 1K tokens |
| `text-davinci-003` | $0.0200 / 1K tokens | $0.0200 / 1K tokens |
| `text-curie-001` | $0.0020 / 1K tokens | $0.0020 / 1K tokens |
| GPT-3.5-Turbo (4K context) | $0.0015 / 1K tokens | $0.002 / 1K tokens |
| GPT-3.5-Turbo (16K context) | $0.003 / 1K tokens | $0.004 / 1K tokens |
| GPT-4 (8K context) | $0.03 / 1K tokens | $0.06 / 1K tokens |
| GPT-4 (16K context) | $0.06 / 1K tokens | $0.12 / 1K tokens |

Table 1: Costs of different LLM APIs offered by OpenAI Ope (a)

**Main Contributions:**

1. We propose SELECTLLM: that estimates the output quality of LLMs for document summarization and optimally routes to LLMs, subject quality and cost constraints.

2. We theoretically study the underlying constrained optimization problem and propose polynomial time algorithms for important special cases.

3. We empirically validate our proposed methods on public as well as enterprise datasets. We not only reduce costs by 40-90%, but also improve observed quality by 4-7%.

4. We further report results from a user study to validate our model selection with qualitative human perception.

5. We release the annotated training datasets generated from open-source data for further exploration by the community.

## 2 Related Work

Typically, model and LLM selection problems have been studied as model cascade problems in the literature, where models or APIs are queried sequentially and selectively in an online manner, for example, Chen et al. (2023; 2020); Mamou et al. (2022); Khalili et al. (2022). In particular, FrugalGPT Chen et al. (2023) is an LLM Cascade-based solution that decides the performance of an LLM after getting the API response. They employ both a predictor and an allocator model along with a scoring function to evaluate the responses of different LLMs. However, this approach introduces latency overhead due to its inherently sequential nature. LLM-blender Jiang et al. (2023a) is an ensembling framework that attains higher quality by blending or fusing the output of multiple LLMs, however cost and latency are not considered in this work. Shnitzer et al. (2023) learns a classifier for each model that predicts whether that LLM can be used for a given input task or not and the focus is on out-of-distribution modeling. At the time of submission we became aware of Ding et al. (2024) that classifies queries into easy or hard in a two-model setting, based on which queries are routed to either the smaller model or the larger model. Unlike ours, their work is applicable only to a pair of models. Moreover, we do a rigorous cost, quality, and latency-based optimization after the model output quality prediction to determine the optimal (constrained) trade-off for the model choice.

There have been some works on prompt length reduction Jiang et al. (2023b); Ghalandari et al. (2022) to reduce costs. These works are complementary to our approach. Also, some works have explored caching to reduce costs and latencyBang (2023); Zhu et al. (2024); Mohandoss (2024), but these are only applicable where user queries for different users come from the same distribution.

## 3 Problem Description

In this section, we describe the problem and framework. We consider the document summarization task. When a new context (query) arrives for summarization, we need to route it to a model such that the quality of the summary is high, while the cost remains within a user-specified budget, and latency is also within a threshold.

Let us first discuss the setting and notations. We have access to a set of $K$ models (LLMs), either through local deployment or through APIs: $\mathcal{M} = \{M_1, M_2, \ldots, M_K\}$. Whenever a model $M_i$ is queried, it incurs a cost and

latency (we will define these in further detail in Section 5). For defining the problem statement, for the moment, let us assume that there is a quantitative scoring function $S$ for evaluating the quality of summaries. Specifically, $S_{i,j}$ denotes the quality score for the summary generated for the $i^{th}$ context by the $j^{th}$ model.

There are two main questions here that we aim to study. First, how can we estimate the quality score vector $\mathbf{S}_i$ for the $i^{th}$ context, where $\mathbf{S_i} = \{S_{i,j} \forall M_j \in \mathcal{M}\}$ without invoking the LLMs? Second, given quality score estimates, how do we determine the optimal allocation of contexts to models?

We now state these two problems formally.

**Problem 1: Quality Estimation** For a given text $T$, let the true quality score vector be $\mathbf{S}$. Assuming the context length of $T$ is $d$, we want to learn a function $LM_\Theta : R^d \rightarrow R^K$ which will act as an approximation to the true quality scoring function. In other words, $LM_\Theta(T) = \hat{\mathbf{S}}$, such that $\hat{\mathbf{S}} \approx \mathbf{S}$.

**Problem 2: Optimal Allocation** We are given a collection of contexts (or, prompts) $\mathcal{T} : \{T_i\}$, and a set of models $\mathcal{M} : \{M_j\}$. We are also given the estimated quality scores $\{S_{i,j}\}$ for each $\{T_i, M_j\}$ pair. Furthermore, we are aware of the usage costs of the models and we are given a budget for the usage costs as well as a latency threshold to adhere to. Our goal is to find an optimal allocation of contexts to models such that the estimated quality scores are maximized in aggregation subject to cost and latency constraints.

## 4 Problem 1: Quality Estimation

In this section, we study the problem of estimating the output quality of LLMs for various prompts for summarization tasks. In general, the LLM performance can vary with the task and the context. For example, it can vary with the domain of the text and, the format of the text among others. We empirically observed that while GPT-3.5-Turbo is better at summarizing data in tabular format compared to `text-davinci-003`, the latter summarizes bulleted list points better than the former. Hence, a key (empirical) insight is that there might not be a clear hierarchy of the models in terms of their performance (also noted by Chen et al. (2023)), and estimating the LLM response quality is non-trivial. Existing works such as FrugalGPT Chen et al. (2023) *need to invoke the LLMs at run time* in order to evaluate their performance. However, that is counterproductive to our use case and objectives, as querying each LLM separately will not only *increase the costs a lot, but will also result in high latency*. We have proposed a model to predict the output quality of LLMs with high fidelity.

### 4.1 Choice of Quality Metric

To assess the output quality, firstly, it is essential to establish a quantitative metric for evaluation. In scenarios like multiple choice question answering and Natural Language Inference (NLI) tasks related to document processing, we can rely on accuracy and NLI metrics, respectively, to quantitatively measure performance. However, in cases where the task is inherently more subjective and qualitative, like text summarization, selecting an appropriate evaluation metric becomes less straightforward. In the literature, different scores that have been used for this purpose include variants of ROUGE scores, such as ROUGE-1, ROUGE-2, and ROUGE-L Lin (2004) as also BLEU metrics Papineni et al. (2002) and METEOR scores Banerjee & Lavie (2005). These metrics however don't have a deep understanding of the semantics or context of the language as they are based on n-gram matching, which can lead to inaccuracies, especially in tasks that require nuanced or context-aware language generation. BERTScore Zhang et al. (2019) and BARTScore Yuan et al. (2021) were shown to capture semantic notions of generated text better and highly correlated with human judgment, hence these are more suitable for quantitative evaluation of the qualitative perception of the summary. We believe that it is an easier task to allocate task to summarization models than to actually summarize the text. Our work is focused on generating concise summaries of 2-3 lines, so a context window of more than 512 tokens is unnecessary for evaluating the produced summaries. Additionally, we conducted an extensive survey to compare human opinion alignment with BERTScore and BARTScore. Based on the results and considering the lower computational resource requirements, we ultimately chose BERTScore as our quality evaluation metric.

### 4.2 Proposed Model: Bert-based Score Predictor

Figure 1 illustrates our proposed model framework: Bert-based Score Predictor. It takes as input a given piece of text that is to be summarized and generates a quality score for each model in the cascade. These scores represent how well

each model would summarize the text compared to a gold standard, which can be human summaries or summaries generated by a powerful LLM. We use a Language Model: Bert as our backbone, and on top of this, we add a regressor head for the final prediction. Additionally, this regressor head incorporates Layer Norm between successive layers. From extensive experimentation, we found that using GELU activation yielded the most favorable results.

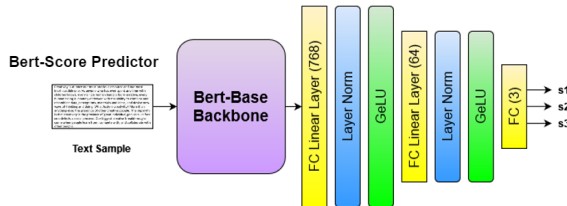

Figure 1: Bert-based Score Predictor

## 4.3 Generating Ground Truth

For training the above model we need to generate the ground truth as a first step. We start by annotating datasets with reference quality scores. These scores are determined by first obtaining the gold standard summary by querying the most advanced large language models that we have access to (GPT-4 or GPT-3.5-Turbo). Subsequently, we query each model within our cascade to generate candidate summaries, based on which the quality scores are calculated. We have curated two distinct datasets, Dataset-I and Dataset-II, following this methodology. Dataset-I comprises approximately 1000 text sections extracted from real-world PDF documents obtained from Adobe Inc. Gold summaries for this dataset were generated using GPT-4, with the cascade of models including `text-davinci-003`, `text-curie-001`, and GPT-3.5-Turbo. On the other hand, Dataset-II consists of around 3000 text samples from three sources which are the bigpatent Sharma et al. (2019), samsum Gliwa et al. (2019), and wiki-bio Lebret et al. (2016) datasets.We selected these three datasets because they represent different types of content typically found in documents. bigpatent contains U.S. patent documents, samsum includes collections of dialogues, and wiki-bio provides structured data, such as tables. Additionally, we chose not to use the gold standard summaries provided in these datasets because there is a lack of continuity between the summaries across the different datasets, and they vary in length. Our focus is on generating concise, 2-3 line summaries for general use cases. Each data point in Dataset-II[1] was annotated with a quality score (BERTScore in this case), using GPT-3.5-Turbo's summaries as the reference gold standard. For this dataset, the cascade included `text-davinci-003`, `text-curie-001`, and `vicuna-13b`.

## 4.4 Loss Function

Using the quality scores as ground truth for each input text $d_i$, the Bert-based Score Predictor generates $K$ scores where $K$ is the number of LLMs considered in the cascade ($K = 3$, in our case). Let $\hat{\mathbf{y}}^i \in \mathbb{R}^K \geq 0$ denote the vector of the actual quality scores incurred on the $K$ models for section $d_i$ and $\hat{\mathbf{y}}^i \in \mathbb{R}^K \geq 0$ is the predicted vector. For a pair of distinct models $k_p$ and $k_q$, let $\Delta^i_{k_p,k_q} = \mathbf{y}^i(k_p) - \mathbf{y}^i(k_q)$ and $\hat{\Delta}^i_{k_p,k_q} = \hat{\mathbf{y}}^i(k_p) - \hat{\mathbf{y}}^i(k_q)$. For a batch size $n'$, the loss is computed as a combination of :
**1. Mean Square Error (MSE) Loss:**

$$\mathcal{L}_{MSE} = \frac{1}{n'} \sum_{i \in [n']} ||\mathbf{y}^i - \hat{\mathbf{y}}^i||^2 \tag{1}$$

**2. Pairwise difference Loss:**

---

[1]We will release this annotated dataset to the community

$$\mathcal{L}_{diff} = \frac{1}{n'} \sum_{i \in [n']} \frac{2}{K(K-1)} \tag{2}$$

$$\cdot \left( \sum_{k_p, k_q \in [K], k_p \neq k_q} (\Delta^i_{k_p, k_q} - \hat{\Delta}^i_{k_p, k_q})^2 \right)$$

Hence, our loss function is: $\mathcal{L}_{total} = \alpha \mathcal{L}_{MSE} + \beta \mathcal{L}_{diff}$, where $\mathcal{L}_{diff}$ was added as a regularizer to the MSE loss to help reinforce or preserve pairwise trends between models, which becomes important in model selection.

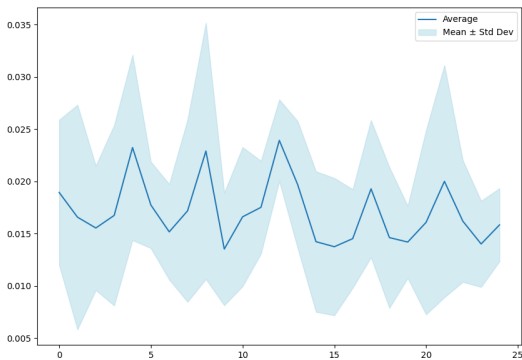

Figure 2: Loss mean and standard deviation of our model across 5 experiments

We utilized the pre-trained `bert-base-uncased` model from Hugging Face[2] as our backbone model. Our regressor head consisted of three linear layers with LayerNorm between each layer and employed the GELU activation function for every layer except the last one. This configuration was fine-tuned on our datasets. The initial learning rate used was 1e−3, with Adam optimizer, with hyperparameters $\alpha = 1$ and $\beta = 2.4$ and trained on one Nvidia a10g GPU for 10 epochs. We used a 70-30 split for train and test on both the dataset. On Dataset I, the training MSE obtained was 5.8e−3, and the test MSE was 6.5e−3. On Dataset II, the training MSE obtained was 2.7e−3, and the test MSE was 9.5e−3. We also show in figure (2) the mean and standard deviation of the loss across 5 experiments to emphasize the statistical significance of our method. Our model requires only 7GB of memory to run with each epoch taking about 16 seconds to finish with a batch size of 8, emphasizing its small footprint.

## 5 Problem 2: Optimal Allocation

In this section, we discuss the constrained optimization problem of selecting LLMs optimally for each context, such that certain constraints are satisfied. We will formalize the problem and develop algorithms to solve it.

Recall from Section 3 that we have access to a set of $K$ models (LLMs), either through local deployment or through APIs: $\mathcal{M} = \{M_1, M_2, \ldots, M_K\}$. Whenever a model $M_i$ is queried, it incurs the following costs (defined similarly to Chen et al. (2023)):

1. cost per token of the input prompt $C^I_i \geq 0$;

2. cost per token of the output response $C^O_i \geq 0$;

3. a fixed cost of invoking a model[3] $C^F_i \geq 0$.

---

[2] https://huggingface.co/bert-base-uncased

[3] This can be the fixed cost (compute, network I/O, service charge etc.) of calling a particular API or invoking a locally hosted model, which can incur compute charges and/or cluster activation charges.

Then the total cost incurred by an input $P^I$, with corresponding output $P^O$ from model $M_i$ is: $C_i^I \cdot |P^I| + C_i^O \cdot |P^O| + C_i^F$. It is possible that $C_i^I = C_i^O$, and $C_i^F = 0$ for any or all $i \in [K]$.

In addition to the monetary cost, each invocation of a model incurs certain latency. The latency incurred is proportional to the token length of input and output and also depends on the particular choice of API, or the local instantiation of the model. Generally, it has also been observed to be proportional to the model size. Let the latency per unit token length as incurred by model $M_i$ be $L_i$. In many cases, estimated latency caused due to input and output often varies, and hence, for further generality, we assume that the latency per unit input token is $L_i^I$ and latency per unit output token length is $L_i^O$. Therefore, when calling model $M_i$, the total latency experienced by an input of token length $|P_i^I|$ (corresponding to the tokenizer for $M_i$) and corresponding output of length $|P_i^O|$ would be $L_i^I \cdot |P_i^I| + L_i^O \cdot |P_i^O| + \mathcal{N}$, where $\mathcal{N}$ denotes noise (mean 0) in the estimation due to network and system state related stochasticity.

As stated earlier, we mainly focus on document summarization. A document $\mathcal{D}$ can be considered to be set of $n$ sections: $\mathcal{D} = \{d_1, d_2, \ldots, d_n\}$. Each section needs to be summarized to a $p$ line summary, where $p$ is a system-defined (or, user-specified) constant. For a given summary for $d_j$ from LLM $M_i$, (assume) we have a quantitative estimate of the output quality as a score $S_{i,j}$.

Each invocation of a model comes with a specific cost. If the model $M_i$ is chosen for section $d_j$ ($j \in [n]$) from the document $D$, then the cost incurred is given by: $C_{i,j} = C_i^I \cdot |d_{i,j}^I| + C_i^O \cdot |d_{i,j}^O| + C_i^F$. Here, $|d_{i,j}^I|$ denotes the token length of the (input) section $d_j$ corresponding to the tokenizer of $M_i$, and $|d_{i,j}^O|$ denotes the corresponding token length of the output summary for section $d_j$ by model $M_i$. For $p$ length summaries, we estimate the expected output length as $p$ times the average number of words per sentence from each model $M_i$ (as observed empirically). Let this be $|d_{i,j}^{avg}|$. The cost $C_{i,j}$ is therefore estimated as: $C_i^I \cdot |d_{i,j}^I| + C_i^O \cdot |d_{i,j}^{avg}| + C_i^F$.

## 5.1 Budget Aware Optimizer

Let the system imposed monetary budget for the summarization task on the given context or (document D) be $B$. This means that the total cost incurred should be less than the budget imposed $B$. Let us define an indicator variable $x_{i,j}$ which is 1 when model $M_i$ is chosen to summarize section $d_j$, and 0 otherwise. Therefore, the budget constraint is: $\sum_{M_i \in \mathcal{M}} \sum_{d_j \in \mathcal{D}} C_{i,j} \cdot x_{i,j} \leq B$.

Let the required SLA (service level agreement) on the expected latency be $L$. The expected latency for section $d_j$ if routed to model $M_i$: $\ell_{i,j} = L_i^I \cdot |d_{i,j}^I| + L_i^O \cdot |d_{i,j}^{avg}|$. Let us assume that the $K$ models can be called in parallel to each other (and multiple calls would not incur any sequentiality). The constraint is: $\sum_{M_i \in \mathcal{M}} \ell_{i,j} \cdot x_{i,j} \leq L \quad \forall d_j \in \mathcal{D}$.

The goal is to maximize the total expected quality of summaries generated for all the sections through the respective models chosen for routing. Therefore, the objective is:
Maximize $\quad \sum_{d_j \in \mathcal{D}} \sum_{M_i \in \mathcal{M}} S_{i,j} \cdot x_{i,j}$.
We further need to add a constraint $\sum_{M_i \in \mathcal{M}} x_{i,j} = 1$ for all $d_j \in \mathcal{D}$ to ensure that every section is summarized by one model. The integer linear program for this problem, that we denote by BUDGET-OPT is given next in Equation 3.

$$
\begin{aligned}
\text{Maximize} \quad & \sum_{d_j \in \mathcal{D}} \sum_{M_i \in \mathcal{M}} S_{i,j} \cdot x_{i,j} & (3) \\
\text{subject to} \quad & \sum_{M_i \in \mathcal{M}} \sum_{d_j \in \mathcal{D}} C_{i,j} \cdot x_{i,j} \leq B, \\
& \sum_{M_i \in \mathcal{M}} \ell_{i,j} \cdot x_{i,j} \leq L \quad \forall d_j \in \mathcal{D} \\
& \sum_{M_i \in \mathcal{M}} x_{i,j} = 1 \quad \forall d_j \in \mathcal{D}, \\
& x_{i,j} \in \{0, 1\} \quad \forall d_j \in \mathcal{D}, \forall M_i \in \mathcal{M}
\end{aligned}
$$

We next study the hardness of BUDGET-OPT even under relaxed latency constraints.

**Theorem 1.** BUDGET-OPT *is* NP-HARD.

*Proof.* We show BUDGET-OPT is NP-HARD from KNAPSACK problem. Due to space limitations, further details are provided in Appendix. □

Since BUDGET-OPT is NP-HARD, we relax it to a linear program, where we allow $0 \leq x_{i,j} \leq 1$ in place of the integrality requirement. We consider the latency relaxed version[4]. For obtaining the final allocation, we use the following simple rounding rule (breaking ties by choosing the lower cost model): $\hat{x}_{i,j} = 1$ if $x_{i,j} \geq x_{i',j} \forall i' \in [K], 0$ otherwise. We empirically find that the above rounding violates budget by $< 0.2\%$.

## 5.2 Quality Aware Cost Minimizer

Here we study theoretically another practically important variant of the problem COST-MIN where a quality threshold $Q$ must be maintained *at a per instance level* while minimizing the total costs. The corresponding integer linear program is:

$$\text{Minimize} \quad \sum_{d_j \in \mathcal{D}} \sum_{M_i \in \mathcal{M}} C_{i,j} \cdot x_{i,j} \tag{4}$$

$$\text{subject to} \quad \sum_{M_i \in \mathcal{M}} S_{i,j} \cdot x_{i,j} \geq Q \quad \forall d_j \in \mathcal{D},$$

$$\sum_{M_i \in \mathcal{M}} \ell_{i,j} \cdot x_{i,j} \leq L \quad \forall d_j \in \mathcal{D}$$

$$\sum_{M_i \in \mathcal{M}} x_{i,j} = 1 \quad \forall d_j \in \mathcal{D},$$

$$x_{i,j} \in \{0, 1\} \quad \forall d_j \in \mathcal{D}, \forall M_i \in \mathcal{M}$$

**Theorem 2.** COST-MIN *is* NP-HARD.

*Proof.* We prove this by a reduction from PARTITION. Further details are provided in Appendix. □

### 5.2.1 Polynomial Special Cases

For two special cases, COST-MIN admits polynomial time algorithms.

**Theorem 3.** *In the absence of latency constraints, an $O(K)$ greedy algorithm gives the optimal solution to* COST-MIN.

*Proof.* We show that a greedy algorithm is optimal in this case. Further details are in the Appendix. □

**Theorem 4.** *When all the sections are equal in length in terms of tokens, then* COST-MIN *admits a polynomial time solution.*

*Proof.* This problem can be modeled as a minimum cost maximum flow problem and as a result, admits a polynomial time optimal solution by the Bellman-Ford algorithm. Further details are provided in Appendix. □

---

[4]This holds when the maximum estimated latency is less than the threshold for any model and text pair, which is a practical scenario, especially for generating short summaries.

| Method | Cost (1e-3 $) | Allocation GPT3.5/Davinci/Curie | Avg. BERTScore |
|---|---|---|---|
| Only `text-davinci-003` | 3549.71 | [0.00, 1.00, 0.00] | 0.746 |
| Only GPT-3.5-Turbo | 709.94 | [1.00, 0.00, 0.00] | 0.761 |
| SELECTLLM (B = 370) | 370.01 | [0.16, 0.00, 0.84] | 0.708 |
| Random (B = 370) | 389.39 | [0.16, 0.00, 0.84] | 0.693 |
| SELECTLLM (B = 550) | **550.12** | [0.79, 0.03, 0.18] | **0.770** |
| Random (B = 550) | 603.77 | [0.77, 0.07, 0.15] | 0.748 |
| SELECTLLM (B=1200) | 1201.01 | [0.62, 0.27, 0.11] | **0.782** |
| Random (B = 1200) | 1378.02 | [0.62, 0.27, 0.11] | 0.748 |

Table 2: Results on Dataset I. We have compared SELECTLLM with three baselines: i) Only `text-davinci-003`, ii) Only GPT-3.5-Turbo, iii) Proportional: Defined in Section 6. Costs are estimated as per OpenAI pricing.

## 6 Experimental Results

To evaluate our approach, we have performed several experiments. We considered the document summarization task, where each text sample needs to be summarized in to 2 line summaries (the prompt to the LLM specified this task with this desired length). For each text sample, we predicted the quality scores corresponding to each model in the set of choices as compared to the Gold standard (latency of predictor model in milliseconds). Then we solved the fractional Linear Program BUDGET-OPT (one time for a dataset, latency in milliseconds) and rounded the optimal fractional solution to obtain integral allocations for each sample to one of the model choices. We report the aggregated quality scores along with the total costs incurred for different values of the budget.

In the first set of experiments, we used Dataset-I. The model choices were GPT-3.5-Turbo, `text-davinci-003`, `text-curie-001`, and the gold summaries were generated using GPT-4. We compare our approach against the following baselines: i) Only `text-davinci-003`, ii) Only GPT-3.5-Turbo, and iii) Proportional Allocation, described next.

**Proportional Allocation:** Consider the optimal fractional solution of the LP. We aggregate the total allocation to each model. For LLM $M_i$, let the total allocation to $M_i$ by Budget-Opt is: $\sum_{d_j \in \mathcal{D}} x_{i,j}$. Let us call this $X_i$. Now we normalize $X_i$ to $X_i'$ as follows $X_i' = \frac{X_i}{\sum_{M_k \in \mathcal{M}} X_k}$. This gives us a probability distribution across the models, where the $X_k'$ can be considered to be the probability of choosing model $M_k$. Hence, in this baseline, for each input, we choose a model $M_k$ with probability $X_k'$. Table 2 lists the results on Dataset-I along with the allocation vectors for each model. The metrics of interest are: i) **Total Cost** incurred, and ii) the **Average BERTScore** of the generated summaries with respect to gold.

SELECTLLM performs significantly better than all the baselines compared. For Dataset I, we get a **84.50%** cost reduction and **3.2%** quality improvement over the "Only `text-davinci-003`" baseline and **22.55%** cost reduction and **1.2%** quality improvement over the "Only GPT-3.5-Turbo" baseline. We can see that the allocation vector for our solution for a budget $B$ is the same as that in the Proportional Allocation baseline given in the adjacent row. However, since the choice of model is not optimized here, both the cost is higher and BERTScore is lower. We have not compared with "Only `text-curie-001`" baseline here as the quality scores on average were quite low. However, including both Da-Vinci and Curie as options in the model choice for SELECTLLM helps to both **lower costs** as well as achieve **higher quality scores** than all the baselines operating at a comparable or higher cost, because of the mathematically optimal trade-off and near correct estimation of scores.

Another interesting observation is that BUDGET-OPT shows diminishing returns with increasing budget. Figure 3 shows the estimated quality scores with increasing budget on Dataset-I. We can see that while initially, it increases rapidly, it saturates at $\approx 0.78$. Also, note that the violation of the budget constraints by the rounding process was observed to be $\leq 0.2\%$, hence, we can approximate the costs by the corresponding budget values.

We perform a similar experiment on Dataset II. In this case, the model choices were `text-davinci-003`, `vicuna-13b` and `text-curie-001` and the gold summaries were generated using GPT-3.5-Turbo. We compare our approach against the following baselines: i) Only `text-davinci-003`, ii) Only `text-curie-001`, iii) Only Vicuna, and, as defined before, iv) Proportional allocation. We get **90%** cost reduction with no degradation in

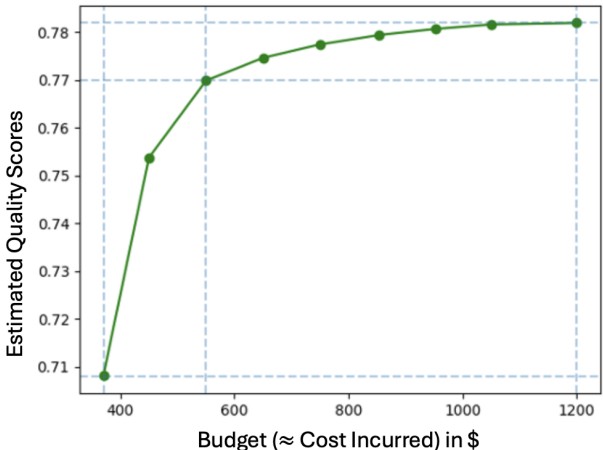

Figure 3: Plot showing the optimal (aggregated) quality scores across Dataset I with varying Budget (hence, costs incurred (at most 0.2% violation observed empirically of the Budget constraints due to the rounding procedure) of BUDGET-OPT.

| Method | Cost (1e-3 $) | Allocation Davinci/Curie/Vicuna | Avg. BERTScore |
|---|---|---|---|
| Only `text-davinci-003` | 8917.28 | [1.00, 0.00, 0.00] | 0.772 |
| Only Curie | 891.728 | [0.00, 1.00, 0.00] | 0.721 |
| Only Vicuna | 234.8 | [0.00, 0.00, 1.00] | 0.686 |
| SELECTLLM (B = 500) | **500.0038** | [0.072, 0.442, 0.486] | **0.751** |
| Proporional (B = 500) | 1151.49 | [0.072, 0.442, 0.486] | 0.722 |
| SELECTLLM (B = 891) | **891.08** | [0.196, 0.487, 0.317] | **0.773** |
| Proportional (B = 891) | 2195.73 | [0.196, 0.487, 0.317] | 0.718 |
| SELECTLLM (B=1500) | 1493.99 | [0.349, 0.495, 0.156] | **0.786** |
| Proportional (B=1500) | 3681.33 | [0.349, 0.495, 0.156] | 0.718 |

Table 3: Results on Dataset II. Here, we have compared SELECTLLM with four different baselines: i) Only `text-davinci-003`, ii) Only `text-curie-001`, iii) Only `vicuna-13b`, iv) Proportional: Same as in DatasetI. Cost of OpenAI models were calculated as per OpenAI pricing. Vicuna was self-hosted and its cost is estimated by the compute cost per hour of the renting hardware (GPU) and token throughput.

quality compared to DaVinci baseline, and similar cost but **7.21%** quality improvement over the Curie baseline. Also, at a lower budget, we achieve a cost reduction of **43.9%** and quality improvement of **4.16%** over only Curie baseline.

We have also compared it with an LLM Cascade baseline inspired by FrugalGPT. FrugalGPT calls three LLM APIs sequentially to generate the query result. If the response from LLM APIs exceeds a certain performance threshold, no further API calls are made. We use two different ordering of APIs for our experiments. The first ordering is `text-curie-001`, GPT-3.5-Turbo and `text-davinci-003` (FrugalGPT davinci) and for the second ordering, we swap GPT-3.5-Turbo and `text-davinci-003` (FrugalGPT 3.5). Figure 4 shows the plot of cost vs Avg. BERTScore for different approaches. Our method achieves the same BERTScores as the FrugalGPT-inspired baselines at significantly lower cost.

## 6.1 User Study

We also conducted a user survey to see how well our predictor module (which is based on BERTscore) aligns with human preferences. Participants were presented with a piece of text and two summaries generated by different LLMs. They were asked to choose which summary they preferred, with the following options: Model A, Model B, both summaries are adequate, or neither summary is adequate. We clarified to the participants that "adequate" means the summary aligns with how they would have summarized the given text. The LLMs used here were `text-davinci-003` and `text-curie-001`. The participants were not made aware of which summary is generated by which model. Out of the 10 texts shown to users, half of the texts were where our predictor module predicted that curie (the cheaper

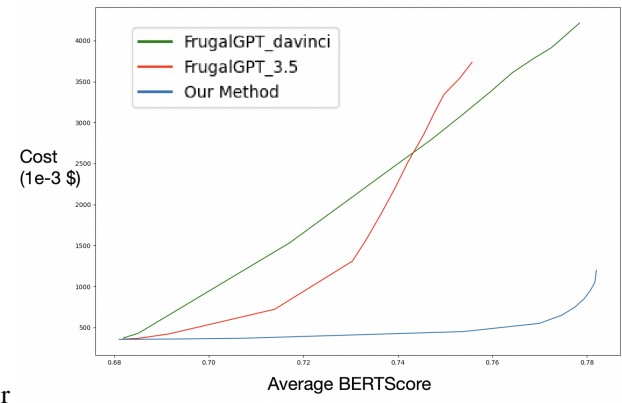

r

Figure 4: Comparison with an LLM Cascade baseline inspired by FrugalGPT. We achieve the same quality at considerably lower costs and latency (not shown here).

LLM) will be adequate for summarization. 3 of the texts were where our predictor module predicted that davinci would be significantly better than curie, whereas 2 texts were those where there was no significant difference between the predictions (our module predicted both LLMs to perform similarly). We obtained responses from over 50 users (total data points $n > 400$).

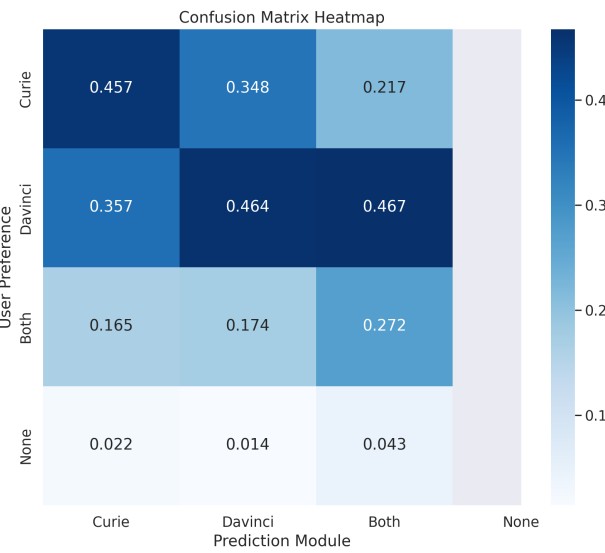

Figure 5: Confusion matrix for user study, we can see that the user preference is highly correlated with the prediction by SE-LECTLLM.

Figure 5 shows the normalized confusion matrix, comparing our prediction module's suggested LLMs (based on just the input text alone) with user preference of final summaries generated. As we can see, there is a strong correlation with human preference when our model predicts either DaVinci or Curie. This means we can effectively predict how users would prefer a model-generated summary when our model predicts a significant gap between the LLMs. When the predicted score gap between LLMs was low (when our model predicted 'both'), we found a low correlation with human preference. Looking at the actual questions, we find that humans strongly preferred 'both' in one of the questions, while preferring DaVinci for the other. This points to the overall hardness of predicting the 'correct' LLM when both models are close in performance. However, when there is a significant performance gap, our module is able to predict it with a high correlation with human preference.

**Latency:** Even though in the experiments, we did not specify latency constraints in BUDGET-OPT, we obtain a $\approx 13\%$ **reduction in total API call wait time** owing to a significant percentage of queries being routed to `text-curie-001`, having lower response time.

## 7   Conclusion and Future Work

We present SELECTLLM: a quality aware framework for reducing costs of LLM usage. We have shown significant cost savings, and comparable quality in most cases, and in some cases, even improvement in quality due to context-based smarter choice of LLMs. We would like to extend our framework to the fully online setting, where the LLM quality estimation can be done contextually in an online manner.

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

# A  Appendix

## A.1  Proof of Theorem 1

*Proof.* Consider a relaxed instance, where there are no latency constraints and there are only 2 models: $M_1$ that has $c > 0$ cost per token and $M_2$ has 0 cost per token. Let us consider an input document $\mathcal{D} = \{d_i\}$, where $d_i$ has a accuracy score $S_{i,1}$ in $M_1$, and $S_{i,2}$ in $M_2$. On model $M_1$, the expected (input + output) token length of $d_i$ is $T_i$ and hence its cost is $c \cdot T_i$. Our goal is to maximize the total quality score of the assignments while maintaining the total cost $\leq B$, where B is the budget.

Let $\mathcal{D}'$ denote the set of document sections where $S_{i,2} \geq S_{i,1}$. Without loss of generality, any optimal solution would assign $\mathcal{D}'$ to $M_2$, as otherwise, we can always swap the assignment and get better or same quality score at a lower cost. Hence, we can remove these from the decision problem.

Let $\mathcal{D}''$ denote $\mathcal{D} \setminus \mathcal{D}'$. Without loss of generality, for each $d_i \in \mathcal{D}''$ let $S_{i,1} = S_{i,2} + \Delta_i$, where $\Delta_i > 0$.

Let the total quality score of the any feasible solution be $S_F$. This consists of scores from sections assigned to $M_2$ as well as $M_1$. Let the sections from $\mathcal{D}''$ assigned to $M_1$ be $\mathcal{D}_1$ and those from $\mathcal{D}''$ assigned to $M_2$ be $\mathcal{D}_2$. Therefore:

$$S_F = \sum_{d_i \in \mathcal{D}'} S_{i,2} + \sum_{d_j \in \mathcal{D}_1} S_{j,2} + \Delta_j + \sum_{d_k \in \mathcal{D}_2} S_{k,2} \qquad (5)$$
$$= \sum_{d_i \in \mathcal{D}} S_{i,2} + \sum_{d_j \in \mathcal{D}_1} \Delta_j = S_2 + \sum_{d_j \in \mathcal{D}_1} \Delta_j$$

where $S_2$ is constant, as defined by the input instance. An optimal solution would be maximizing the second component of the above in a feasible way. Therefore, the optimization problem reduces to the following: finding the subset of sections $d_i$ from $\mathcal{D}''$, each of cost $c \cdot T_i$, that can be feasibly assigned to $M_1$, without violating the budget $B$, while maximizing the quality score (sum of $\Delta_i$'s) of the assigned sections. This exactly equivalent to 0-1 KNAPSACK. Formally, we are given an instance of 0-1 KNAPSACK with $n$ items, each item has value $v_i$ and weight $w_i$, and a knapsack with capacity $C$. We create an instance of our problem with $n$ sections. For each section $i$, we let $S_{i,2} = z_i$ where $z_i \geq 0$ is a random number and $\Delta_i = v_i$. We choose the cost of $d_i$ as $T_i = \frac{w_i}{c}$ and budget $B = C$. We can see that if there exists a feasible solution of total value $V$ in knapsack, that implies that BUDGET-OPT on the created instance has a feasible solution of quality score at least $V + S_2$, where $S_2 = \sum_{i \in [n]} z_i$ (by using the corresponding assignments). Similarly, if our problem has a feasible solution of quality score $Q'$, that implies, that there exists a feasible solution of value at least $Q' - S_2$ for the Knapsack instance. This completes the proof.

$\square$

## A.2 Proof of Theorem 2

*Proof.* For the NP-HARDNESS proof, let us consider a simplified version of the problem where there are only 2 models, each with 0 cost and the quality constraints are satisfied for both the models for both the sections. Let us consider the feasibility version of the problem. Specifically, the decision question is whether there exists an assignment of the sections to the 2 models such that the latency constraints are satisfied for each model. We reduce from PARTITION for this problem. Given an instance of PARTITION with $n$ elements of size $\{a_1, a_2, \ldots, a_n\}$, such that $\sum_{i \in [n]} a_i = 2B$, we need to find if there exists a partition of the elements such that each partition sums to $B$. We create an instance of COST-MIN with 2 models, and $n$ sections. We choose a random number $z < \min_{i \in [n]} \{a_i\}$. We set the output size for every section to be $z$, and the input size of section $a_i - z$, therefore, the total token size of $d_i$ is $a_i$. Let the latency coefficient $\ell_j$ for each model $M_j$ be equal to $\ell$. The latency threshold for either model is set to be $L = \ell B$. The decision question is whether there exists a latency feasible solution for COST-MIN in the given instance. We can see that a YES instance for PARTITION implies a YES instance for COST-MIN, by simply assigning the document sections corresponding to the elements in each partition of total size $B$ to each model. The total latency in each model would therefore be $\ell B = L$. Similarly, a YES instance for COST-MIN would imply a YES instance for PARTITION. We simply take the document sections assigned to each model, and assign the corresponding elements to each partition. The total size of elements in each partition would then be $\frac{L}{\ell} = B$. This completes the proof. $\square$

## A.3 Proof of Theorem 3

*Proof.* For each instance $d_j$, we first find the set of feasible models $\mathcal{F}_j$. These would be the models that satisfy the quality constraints, that is, $M_i \in \mathcal{F}_j$ if and only if $S_{i,j} \geq Q$. This requires $O(nK)$ computations for all $\mathcal{D}$. Then we find the minimum cost model $M' = \arg\min_{M_i \in \mathcal{F}_j} C_i$ for each $d_j$ in $O(K)$ and assign $d_j$ to $M'$. The cost incurred would be minimum. In order to see the proof, let us assume by contradiction, that, the optimal solution deviates from the greedy solution for some section $d_j$ and chooses model $M_j^{opt}$ in place of the greedy choice $M_j$. Clearly, $M_j^{opt}$ must be a feasible model for $d_j$, otherwise, the optimal solution would be violating the quality constraint. Since greedy chose the minimum cost model $M_j$, replacing $M_j^{opt}$ cannot increase the cost of the solution. This is true without loss of generality for any $j$ where the optimal solution is different from the greedy. Hence, the optimal solution can be feasibly converted to the greedy solution without increasing the cost, since there are no latency constraints. This completes the proof. $\square$

## A.4 Proof of Theorem 4

*Proof.* This problem can be modeled as a minimum cost maximum flow problem and as a result admits a polynomial time optimal solution by the Bellman Ford algorithm. The construction is as follows. We construct a directed bipartite graph with the sections as nodes in one partition and the models as the nodes in the other partition. Specifically, we construct a graph $\mathcal{G} = \{\mathcal{V}_1, \mathcal{V}_2, \mathcal{E}\}$, where $\mathcal{V}_1 = \mathcal{D} = \{d_1, d_2, \ldots, d_n\}$, and $\mathcal{V}_2 = \mathcal{M} = \{M_1, M_2, \ldots, M_K\}$, and $\mathcal{E}$ is comprised of feasible directed edges between the nodes in the two partitions. The edges are all directed from the document section nodes to the model nodes. An edge $e = (d_j, M_i)$ (i.e., directed from $d_j$ to $M_i$) exists only if it is feasible, that is, if the assignment meets the estimated quality constraints: $S_{i,j} \geq Q$.

A model $M_i$ can accommodate $N_i = \lfloor \frac{L}{L_i} \rfloor$ tokens while satisfying latency constraints. Let us refer this to as $M_i$'s token capacity. Let the (input + output)[5] size of every section be $d$ in terms of number of tokens. Let us normalize the model capacities as well as by the section sizes by $d$ without loss of generality. Now, the sections have size 1 and the normalized model capacity for $M_i$ is $\hat{N}_i = \lfloor \frac{N_i}{d} \rfloor$. Therefore, we can assign $\hat{N}_i$ document sections to model $M_i$ without violating latency constraints.

Now, we set up a flow problem in this graph. We construct a source node $s$ and a sink node $t$. We construct directed edges from $s$ to each document section $d_j$, and set its capacity 1 and cost as 0. The edges directed from section nodes to model nodes each have capacity 1 and cost corresponding to the model cost. Specifically, an edge $e = (d_j, M_i)$ has capacity 1 and cost $C_{i,j} \cdot d$. We further construct directed nodes from each of the model nodes to the sink $t$. For an edge $e' = (M_i, t)$, the cost is 0 and the capacity is $\hat{N}_i$. Now, for $n$ document sections, we try to send a flow of $n$ from $s$ and $t$ and find the minimum cost maximum flow in this graph. If the problem admits a feasible solution, that is, if there exists a solution such that all document sections can be assigned to one model each without violating quality and latency constraints, then, by integrality of flow and the optimality of min-cost max flow algorithm (one can use Bellman Ford algorithm for this purpose), we will find the minimum cost such assignment. The assignment would be: if an edge $e = (d_j, M_i)$ carries a flow of 1, then document section $d_j$ should be assigned to model $M_i$, otherwise not. On the other hand, if there exists no such feasible solution, then the flow will find the maximum number of feasible assignments at the minimum cost. The complexity is polynomial: $O(|V|^2|E|)$. $\square$

---

[5]The expected output token size is same for all sections by our earlier assumption of $p$ sentence summary. We can simply multiply $p$ by the estimated average number of tokens per sentence as observed through empirical data.

