# OpenReview forum: "Towards Optimal LLM Selection"
_TMLR — Rejected by TMLR_

### Review · Reviewer_9Atz · 2025-02-15

**Summary Of Contributions:**

The paper presents SELECTLLM, a framework for quality-aware, cost-efficient selection among large language models (LLMs) for document summarization. It introduces a novel approach that first predicts the expected output quality of different LLMs using a Bert-based score predictor, and then formulates an optimization problem (with both budget and latency constraints) to select the optimal model for each input section.

**Audience:**

Yes

**Claims And Evidence:**

Yes

**Requested Changes:**

Expand the evaluation to include additional generative tasks beyond document summarization. This would help demonstrate the generalizability of SELECTLLM to other applications such as question answering or translation.
Provide a more detailed analysis of the LP rounding procedure. Discuss its performance and robustness under varying latency and budget constraints, and consider including sensitivity analyses.

**Strengths And Weaknesses:**

The formulation is both timely and practically relevant. As enterprises seek to balance quality with operational costs when using LLMs, this work provides a rigorous framework for making such trade-offs.  The paper integrates theoretical rigor with practical implementation. The NP-hardness results, LP relaxation, and rounding strategies are well-motivated, while the experiments validate that the approach reduces costs by up to 90% while maintaining or even improving quality.

However: The study is focused solely on the document summarization task. It remains an open question how well the approach generalizes to other generative tasks (e.g., question answering or translation) that may have different cost or quality trade-offs. • Some practical assumptions (e.g., fixed cost per token, parallel API calls without additional overhead) might oversimplify real-world scenarios where latency and cost dynamics can be more complex. • While the LP rounding procedure is shown to be effective, more discussion on its robustness under varying system constraints would be valuable. • The user study, although supportive, is based on a moderate sample size. Future work could benefit from a broader study to further validate human alignment. • The paper could improve clarity in some sections—particularly the notation and the explanation of the optimization formulation—so that readers from a broader community can more easily grasp the technical details.

---

### Review · Reviewer_EhME · 2025-02-26

**Summary Of Contributions:**

This paper addresses the high cost of LLMs in large-scale request scenarios by proposing a request scheduling mechanism that routes simpler requests to cheaper model APIs and complex requests to more expensive model APIs. The problem is framed into two key components: output quality estimation and budget allocation. Experiments conducted on GPT-3.5-Turbo, text-davinci-003, and text-curie-001 demonstrate a 40-50% cost reduction while improving performance by 4-7%.

**Audience:**

Yes

**Claims And Evidence:**

No

**Requested Changes:**

1. Add relevant baselines for comparison.
2. Discuss the differences and contributions relative to these baselines.
3. Expand experiments across more tasks and scenarios.
4. Include results on other model types, particularly on-device or weaker models.

**Strengths And Weaknesses:**

Strengths:
1. The paper tackles a crucial real-world problem—dynamic API request scheduling—especially given that most requests do not require the most powerful models.
2. The proposed method effectively reduces overall API costs, as demonstrated by the experiments.

Weaknesses:
1. The paper lacks key baselines, such as [1,2,3], making it difficult to benchmark improvements.
2. The quality estimation approach has been extensively explored in prior work, making its claimed contribution less novel.
3. The experiments are limited to summarization tasks, lacking broader validation.
4. The evaluation is conducted only on OpenAI’s powerful models, leaving uncertainty about its effectiveness for on-device or weaker models.

[1] RouteLLM: An Open-Source Framework for Cost-Effective LLM Routing. ICLR 2025.

[2] TensorOpera Router: A Multi-Model Router for Efficient LLM Inference. EMNLP 2024.

[3] Eagle: Efficient Training-Free Router for Multi-LLM Inference.

---

### Review · Reviewer_fBq6 · 2025-03-01

**Summary Of Contributions:**

The paper introduces methods to optimize the usage costs of Large Language Models (LLMs) for document summarization, achieving cost reductions of 40% to 90% while maintaining high output quality. It presents a model to predict output quality at runtime, ensuring alignment with human preferences, and develops an LP rounding algorithm for optimal query allocation under cost and/or latency constraints. The method is validated through evaluation on publicly available document datasets by considering quality metrics and user preferences obtained through a user study.

**Audience:**

Yes

**Claims And Evidence:**

Yes

**Requested Changes:**

1.There have been a plethora of works on LLM selection in the past year. Here are a few salient ones. Please at least discuss these in the related work survey:

a. [2308.06077] Fly-Swat or Cannon? Cost-Effective Language Model Choice via Meta-Modeling (arxiv.org)

b. [2401.13979] Routoo: Learning to Route to Large Language Models Effectively (arxiv.org)

c. [2403.12031] RouterBench: A Benchmark for Multi-LLM Routing System (arxiv.org)

d. [2404.15153] Expert Router: Orchestrating Efficient Language Model Inference through Prompt Classification (arxiv.org)

e. [2406.18665] RouteLLM: Learning to Route LLMs with Preference Data (arxiv.org)


2. Why do you use GPT-4 to generate gold responses in Dataset 1 and GPT-3.5-Turbo to generate gold responses in Dataset 2?

3. How will you obtain the latency per unit input token ($L_{i}^{I}$) and the latency per unit output token ($L_{i}^{O}$ since this information is not generally publicly available for a model and also depends on the hardware and input size?

4. The proportional allocation obtains a probability of selecting a single model for a given document by normalising the sum of $x_{ij}$'s for a given document. Why not directly use the $x_{ij}$'s as the probability for selecting the model for section $j$ and thus select separate models for each section? These seems more directly comparable to SelectLLM which also selects a different model for each section if I understand correctly.

5. Is proportional allocation the same as Random in Table 2?

6. Why are you plotting "estimated" and not actual quality scoresin Fig 3? If the queries are actually being served by the model selected then we should be able to obtain the actual quality score by comparing the model's response to the ground truth obtained in Section 4.3 right?

7. Do you account for the network latency in your estimate of the API call wait time on Page 11? If not, how can you claim that the reduction in wait time is entirely due to SelectLLM? For e.g. the reduction could also be because the smaller model also had fewer requests waiting in the queue since it is less popular.

**Strengths And Weaknesses:**

Strengths:

1. The paper explicitly encodes cost and latency constraints into the routing decision which is an important practical consideration that has not been made in prior work to the best of my knowledge.

2. SelectLLM generally outperforms the baselines in terms of both BERTscore and cost in the experiments. The authors have also made the effort to validate their approach through an actual user study which is very important, due to the numerous interactive use cases of LLMs, but is rarely done.

Weaknesses:

1. The related work survey is very limited (see requested changes below).

2. SelectLLM is not compared with any of the other works on LLM selection and is only compared against very simple baselines.

3. The approach is only evaluated for summarization even though LLMs are now used for a wide range of tasks and any useful model selection approach would need to be effective for a majority of these tasks.

---

### Decision · Action_Editor_hNdM · 2025-04-07

**Recommendation:** Reject

**Comment:**

The paper presents SELECTLLM, an approach to optimize the selection of LLMs for document summarization, aiming to reduce costs while maintaining high output quality. It estimates the output quality of models without invoking them and uses an LP rounding algorithm for optimal query allocation under cost and latency constraints.

Reviewers raised several concerns: the lack of comparison with recent works, the limited task scope, and assumptions about fixed costs and parallel API calls that may not reflect real-world scenarios. The authors did not provide a response to these concerns.

Given the unresolved issues, I recommend rejecting the paper.

**Audience:**

Yes

**Claims And Evidence:**

The claims made in the submission are partially supported by evidence, as the paper presents promising results in terms of cost reduction and quality improvement. However, the evidence is not entirely convincing due to the limited experimental comparisons.